# Serum-deprivation response of ARPE-19 cells; expression patterns relevant to age-related macular degeneration

**Katherine M. Peterson**[1]*, **Sanghamitra Mishra**[1¤a], **Esther Asaki**[2], **John I. Powell**[2], **Yiwen He**[2¤b], **Alan E. Berger**[3¤c], **Dinusha Rajapakse**[1], **Graeme Wistow**[1]

**1** Molecular Structure and Functional Genomics Section, National Eye Institute, National Institutes of Health, Bethesda, Maryland, United States of America, **2** Office of Intramural Research, Center for Information Technology, National Institutes of Health, Bethesda, Maryland, United States of America, **3** Division of Allergy and Clinical Immunology, Johns Hopkins University School of Medicine, Baltimore, Maryland, United States of America

¤a Current address: Neuberg Center for Genomic Medicine Inc., Apex, North Carolina, United States of America
¤b Current address: Cancer Informatics Branch, National Cancer Institute, National Institutes of Health, Bethesda, Maryland, United States of America
¤c Current address: Department of Surgery, Johns Hopkins University School of Medicine, Baltimore, Maryland, United States of America
* petersonk@nei.nih.gov

**Data Availability Statement:** All RNASeq files are available from the Gene Expression Omnibus database (https://www.ncbi.nlm.nih.gov/geo/) via accession number GSE129964.

## Abstract

ARPE-19 cells are derived from adult human retinal pigment epithelium (RPE). The response of these cells to the stress of serum deprivation mimics some important processes relevant to age-related macular degeneration (AMD). Here we extend the characterization of this response using RNASeq and EGSEA gene set analysis of ARPE-19 cells over nine days of serum deprivation. This experiment confirmed the up-regulation of cholesterol and lipid-associated pathways that increase cholesterol levels in these cells. The gene expression analysis also identified other pathways relevant to AMD progression. There were significant changes in extracellular matrix gene expression, notably a switch from expression of collagen IV, a key component of Bruch's membrane (part of the blood-retina barrier), to expression of a fibrosis-like collagen type I matrix. Changes in the expression profile of the extracellular matrix led to the discovery that amelotin is induced in AMD and is associated with the development of the calcium deposits seen in late-stage geographic atrophy. The transcriptional profiles of other pathways, including inflammation, complement, and coagulation, were also modified, consistent with immune response patterns seen in AMD. As previously noted, the cells resist apoptosis and autophagy but instead initiate a gene expression pattern characteristic of senescence, consistent with the maintenance of barrier function even as other aspects of RPE function are compromised. Other differentially regulated genes were identified that open new avenues for investigation. Our results suggest that ARPE-19 cells maintain significant stress responses characteristic of native RPE that are informative for AMD. As such, they provide a convenient system for discovery and for testing potential therapeutic interventions.

**Funding:** This research was supported by the Intramural Research Program of the NIH, National Eye Institute.

**Competing interests:** The authors have declared that no competing interests exist.

## Introduction

Age-related macular degeneration is a complex disease of the retinal pigment epithelium RPE and the blood-retina barrier that leads to the death of photoreceptors [1, 2]. The blood-retina barrier, consisting of the choroid, Bruch's membrane, and the RPE, is responsible for transporting nutrients, waste products, signaling molecules, and molecules of the visual cycle to and from the photoreceptors while protecting the neural retina from incursions [3, 4]. Genetic association studies have identified alleles for increased risk of and protection from AMD [5–7]. However, the disease can develop even in the absence of known risk variants. AMD is strongly associated with aging and environmental risks such as smoking [8]. AMD's form, severity, and progression result from the particular constellation of genetic and environmental factors [2, 9, 10].

Recent reviews have taken a systems biology approach to understanding AMD, identifying the genes identified by genetic screens and the targets of environmental risks in terms of functional pathways [1, 11–13]. Systems-informed approaches have identified significant pathways associated with the development and progression of AMD. A consensus as to the etiology and progress of the disease is emerging. The systems identified include cholesterol and lipid metabolism and transport; the extracellular matrix (ECM); innate immunity, especially the complement pathway; inflammation; cell death; angiogenesis; and senescence. While these systems can be described and categorized individually, biologically, they interact with and affect one another.

Pathways responsible for cholesterol and lipid metabolism have metabolic, transport, and structural roles in the retina [14, 15]. Cholesterol and lipids are also major components of the deposits and drusen that characterize the onset and progression of AMD [16–18]. Bruch's membrane, a specialized basal extracellular membrane, is the principal constituent of the blood-retina barrier [3, 4] and is responsible for transporting molecules between the vasculature and the outer retina [19, 20]. The components of the ECM determine its consistency and functional capabilities. Complement components of the alternative pathway of innate immunity play a central role in the development of AMD [21, 22], and variants of genes for complement confer the highest risk of AMD. Inflammation is closely associated with the complement pathway, but whether inflammation is the cause or result of complement activation is still unknown. Cell death of photoreceptors and RPE and senescence of the RPE characterize the end stages of AMD [10, 23].

There is substantial evidence that the thinning and collapse of the endothelial choroid layer of the eye is key to the onset of AMD [24–27]. The decrease or loss of communication between the RPE and the choroid starves the RPE of nutrients and serum factors. We have shown that serum deprivation of human RPE-derived ARPE-19 cells induces many responses relevant to AMD [28–30]. Unlike many other cell culture lines, ARPE-19 cells tolerate serum deprivation for as long as two weeks. An earlier study of serum-deprived ARPE-19 cell gene expression using microarrays identified differentially expressed pathways with known associations with AMD, particularly the upregulation of cholesterol and fatty acid pathways [29]. This study increases our understanding of ARPE-19 cells' adaptation to serum deprivation by generating a more comprehensive catalog of the gene expression using RNASeq analysis of an extended time series. Our gene expression analysis uses the systems biology rubric as the framework for the pathway analysis. We confirm that the gene expression patterns instigated by serum deprivation of ARPE-19 cells represent pathways associated with AMD onset and progression. We identified significant changes in cholesterol metabolism, extracellular matrix, complement response, inflammation, senescence, calcification, and circadian rhythm.

## Methods

### Experimental design

RNA sequencing was performed on samples from serum-deprived ARPE-19 cells from days 0, 1, 3, 4, 5, 6, and 9 with three biological replicates for each time point.

### Cell culture

ARPE-19 cells were cultured and serum-deprived, as in Mishra et al. (2016) [29]. Briefly, cells were cultured in DMEM supplemented with 10% FBS to greater than 80% confluence. The day0 sample is serum supplemented. The serum-supplemented media was removed from the cultures and replaced with serum-free DMEM, and cells continued towards confluence. Cell culture continued to the appropriate time point when the RNA was extracted from the cells. Time points = day0, day1, day3, day4, day5, day6, and day9.

S1 Table shows the normalized mean of the gene counts by RNASeq for each of the RPE marker proteins described in Reyes et al. [31]. This culture of ARPE-19 cells expresses the majority of RPE marker genes.

### RNA isolation

RNA was isolated from the confluent cultures on the appropriate day using Trizol$^{TM}$ (#15596–018, Invitrogen) per the manufacturer's instructions. The starting volume of Trizol$^{TM}$ was 1.0ml. The RNA pellet was resuspended in 20μl H$_2$O. RNA concentration was measured using a NanoDrop 100. RNA quality, as judged by RIN, was measured using an Agilent BioAnalyzer 2100 and RNA Nano Chips. All the samples had RIN scores greater than 7.0.

### Fluorescent IHC of human retina

Normal and AMD-affected human eyes of donors aged between 65 and 98 years were obtained from the National Disease Research Interchange (NDRI) (Philadelphia, PA). Samples were obtained within 6 to 14 hours of death.

For sectioning, the eyes were fixed with formalin, washed in PBS, and cryoprotected through a series of 5%, 10%, and 20% sucrose in PBS. Eyes were cut and sectioned through the macula. Sections were incubated in ICC buffer (0.5% BSA, 0.2% Tween-20, 0.05% sodium azide, in PBS, pH 7.3) for one hour at room temperature. Sections were incubated with Anti-Collagen I and Anti-Collagen IV antibodies (Catalog# PA5-29569 and # MA1-22148, Thermo Fisher Scientific, Waltham, MA) diluted 1:100 in ICC buffer and Biotinylated Peanut Agglutinin (PNA) (Vector Laboratories, Burlingame, CA) diluted 1: 200 overnight at 4˚C. Following thorough washes with ICC buffer, the slides were incubated with secondary antibodies anti-rabbit Alexa 488, anti-mouse Alexa 568 (Thermo Fisher Scientific, Waltham, MA) diluted 1: 200, and streptavidin conjugated with Alexa 633 (Vector labs, Burlingame, CA) diluted 1300 in PBS for one hour at room temperature. Sections were washed extensively with ICC buffer, mounted in Prolong Gold + DAPI (Molecular Probes, Eugene, OR), and examined using a confocal Zeiss LSM 880 with Airyscan microscope (Zeiss, USA). The results are typical of results obtained from three eyes of each phenotype, normal and AMD.

### ß-galactosidase staining

Confluent cultures of ARPE-19 cells were serum-deprived for 0 or 9 days. The cells were fixed and stained to detect β-galactosidase activity using the Senescence Detection Kit from Abcam (ab65351, Abcam, www.abcam.com).

## Sequencing

The libraries of the RNA samples were constructed and sequenced by the NIH Intramural Sequencing Center, NISC, using the Illumina HiSeq platform.

The sequencing data is available from the NCBI GEO database: https://www.ncbi.nlm.nih.gov/geo/query/acc.cgi?acc=GSE129964.

## Sequence analysis

The sequence files were aligned using Novoalign (v. 3.02.00) against the UCSF hg19 refFlat. Counts were determined using HTseq (v. 0.61) using the mode intersection-strict. DESeq (v. 1.12.1) with the default settings on R(v. 3.0.1) was used to identify differentially expressed genes [32, 33]. DESeq uses a negative binomial statistical model for its analysis. Genes which had negligible expression for a given contrast were excluded from the analysis of that contrast.

Result sets used for gene set analysis or data display are either count matrices for the three biological replicates from each time point or lists of fold change ratios for each time point (where fold change equals the mean normalized counts at the time point divided by mean day 0 normalized counts for ratios greater than or equal to one, and mean day 0 normalized counts divided by the mean normalized counts at the time point times -1 for ratios less than one; commonly referred to as the signed fold change (SFC)).

## EGSEA

To identify the biological pathways that constitute the adaptive response of ARPE-19 cells to serum deprivation, we used the gene set enrichment algorithm EGSEA (ensemble of gene set enrichment analyses) [34]. Gene set enrichment analysis identifies relevant pathways by identifying statistically overrepresented pathways from the differentially expressed gene list. EGSEA combines the results from multiple gene set enrichment algorithms into combined gene set enrichment and rank scores. Combining algorithms increases statistical power and decreases false positive and false negative errors [34]. EGSEA allows users to interrogate multiple selected gene set collections.

We used the EGSEA R package available from Bioconductor (http://www.bioconductor.org/packages/EGSEA) v1.2.0. The RNA-Seq count data was formatted as a matrix for each contrast of serum-deprived sample time point to day0. We set the EGSEA algorithm to combine 10 GSE algorithms (Camera, Gage, Gsva, Ora, Globaltest, Padog, Plage, Roast, Safe, Ssgsea) and ran it four times on each matrix specifying one of the four gene set collections: MSigDB Hallmark, the C2 gene sets of the MSigDB, GeneSetDB, or the Gene Ontology (GO) as the annotation set [35–38]. The results returned by EGSEA include an overall rank score, P-value, and P-values adjusted by a multiple test correction (Benjamini-Hochberg FDR) for each gene set (S3 Table).

We selected the EGSEA-returned gene sets with adjusted P-values less than 0.05. For the Hallmark gene set collection, we used the adjusted P-value less than 0.05 cutoff or all 50 Hallmark gene sets as appropriate for the discussion. The Hallmark gene set collection's size (50 sets) and composition (most sets contain 100–200 genes) make it especially amenable to data visualization [37].

## Statistical analysis

Statistical analysis procedures were carried out within the DESeq and EGSEA software using the settings noted above.

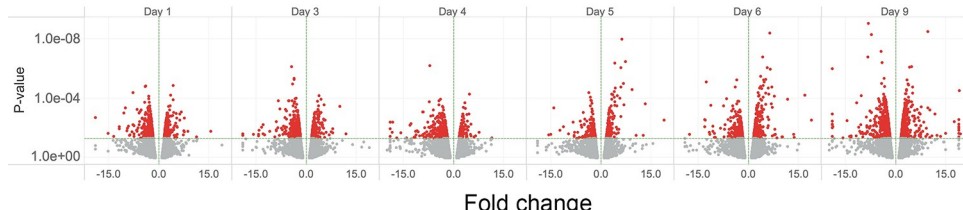

**Fig 1. The small-multiple display of volcano plots.** Scatter plots of RNASeq DESeq results, P-value vs. Fold Change (volcano plots), for each sample day arranged chronologically. Each point represents a gene. The Fold change plotted along the X-axis is the Signed Fold Change (SFC) defined in the Sequence Analysis section above. For readability of the X-axis, SFC values less than -20 and greater than 20 are truncated to -20 or 20. The P-value is plotted on the Y-axis. Color represents the P-value; red points have P-values ≤ 0.05, and grey points have P-values > 0.05. The X-axis reference line is SFC = 0. The Y-axis reference line is P-value = 0.05. While the Y-axis is rendered in log scale, the actual P-value labels are displayed along the axis. Fig 2B is an extended interactive version of Fig 1, displaying the data with gene and measure details.

## Annotation database

The results from the sequence analysis are organized and stored in a MySQL database. We downloaded annotation information from EntrezGene (NCBI, NIH), the Molecular Signatures Database (MSigDB) (Broad Institute and UCSD), and the GeneSetDB. The sequence data and the annotations are linked using the *EntrezID* (number) as the primary key. The amount of annotation associated with each gene is highly variable. At the minimum, each transcript identified in this study has an *EntrezID*, gene name, and symbol mined from the NCBI. Our database includes the EntrezGene summary from NCBI, when available. We mined the Broad Molecular Signatures Database (MSigDB) [36] for the gene set collections from the Hallmark gene sets [37]; the C2 category, which contains the BioCarta, KEGG, and Reactome gene sets [39, 40]; and the C5 category which, includes a modified Gene Ontology (GO), including the Cellular Component (CC), Biological Process (BP), and Molecular Function (MF) sub-categories. See Fig 1 for details of Gene Sets and Gene Set Collections.

## Visualization

We explored and interpreted the results of the RNA-Seq and EGSEA analyses using graphs and dashboards made with Tableau Desktop Software (version 2023.3). Text files generated from database queries joined or blended as necessary using Tableau Prep Software (version 2023.3) served as Tableau data sources. The annotations used for this study are reusable and extensible. They are also portable; the annotation file used in this work can be easily downloaded from the Tableau Workbook and integrated with another quantitative dataset. Tableau interactive figures can be viewed on Tableau Public (see S1 Fig).

## Results and discussion

We serum-deprived ARPE-19 cells for nine days, collected samples for RNASeq on days 0, 1, 3, 4, 5, 6, and 9, as previously described [29], and performed RNASeq analysis. RNASeq identified 16,414 distinct genes during the time course (Fig 1, see Fig 2, S2 Table). Using the R package DESeq to identify differentially expressed genes (DEGs), we detected 2469 genes that met the P-value ≤ 0.05 threshold (see Figs 1 and 2, S2 Table). We performed gene set analysis using the EGSEA R package. Gene set analyses rely on gene annotations to summarize and group gene expression data to identify pathways or functions altered by the experimental conditions. For annotations, we used the Molecular Signatures Database (MSigDB) from the Broad Institute [36] and the Gene Set Database (GSdb) [35] (see Fig 1). The EGSEA procedure

## A.

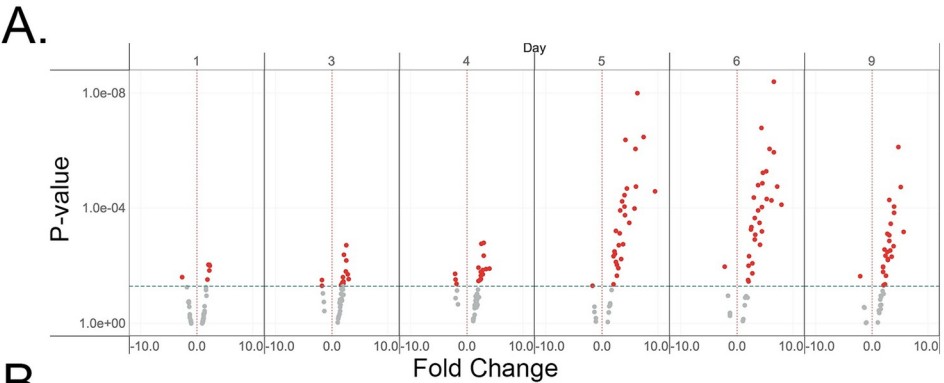

## B.

### 15 Cholesterol Gene Sets identified by EGSEA

EHMN_SQUALENE_AND_CHOLESTEROL_BIOSYNTHESIS

HALLMARK_CHOLESTEROL_HOMEOSTASIS

HUMANCYC_CHOLESTEROL_BIOSYNTHESIS_I

HUMANCYC_CHOLESTEROL_BIOSYNTHESIS_II_(VIA_24,25-DIHYDROLANOSTEROL)

HUMANCYC_CHOLESTEROL_BIOSYNTHESIS_III_(VIA_DESMOSTEROL)

HUMANCYC_MEVALONATE_PATHWAY_I

HUMANCYC_SUPERPATHWAY_OF_CHOLESTEROL_BIOSYNTHESIS

HUMANCYC_SUPERPATHWAY_OF_GERANYLGERANYLDIPHOSPHATE_BIOSYNTHESIS_...

INOH_STEROIDS_METABOLISM

KEGG_STEROID_BIOSYNTHESIS

KEGG_TERPENOID_BACKBONE_BIOSYNTHESIS

REACTOME_CHOLESTEROL_BIOSYNTHESIS

SMPDB_STEROID_BIOSYNTHESIS

WIKIPATHWAYS_CHOLESTEROL_BIOSYNTHESIS(WP197)

WIKIPATHWAYS_STATIN_PATHWAY(WP430)

### 40 Unique Genes

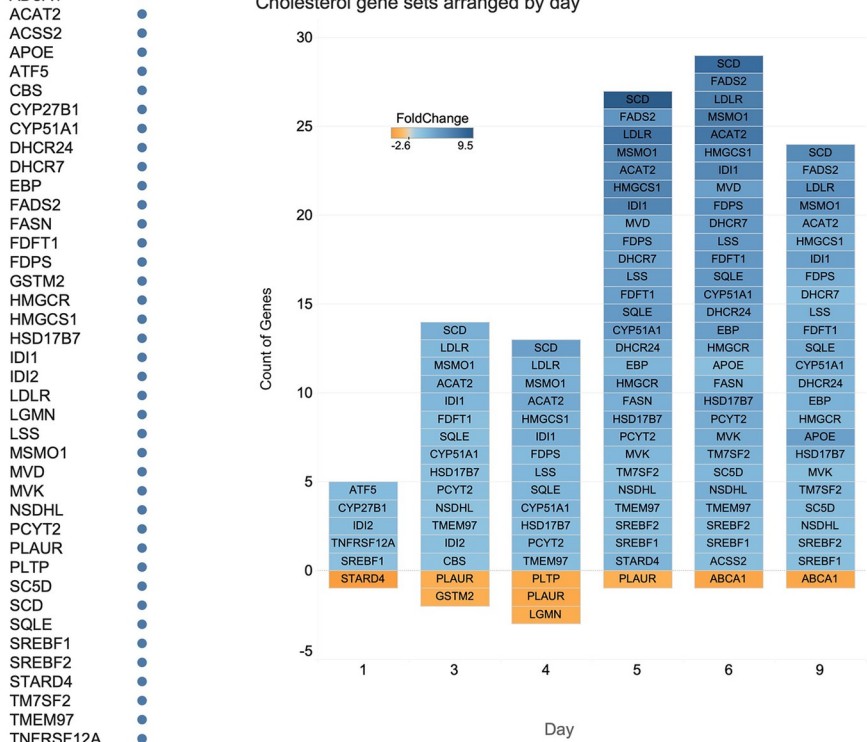

**Fig 2. Significant cholesterol genes arranged by day.** A. Volcano plots of genes identified as Cholesterol Metabolism Genes. Scatter plots of the RNASeq DESeq results, P-value vs. Fold Change for each sample day, arranged chronologically. Each point represents a gene. The Fold Change plotted on the X-axis is the SFC. The P-value is plotted on the Y-axis. Color represents the P-value; red points have P-values ≤ 0.05, and grey points have P-values > 0.05. The X-axis reference line is SFC = 0. The Y-axis reference line is P-value = 0.05. While the Y-axis is rendered in log scale, the actual P-value labels are given along the axis. B. The 15 gene sets identified as significant by EGSEA for Cholesterol contain 40 unique genes. Tiles represent the genes for each sample day when the P-value is ≤ 0.05. Genes with increased expression are blue; genes with decreased expression are orange; the depth of color reflects the magnitude of the increase or decrease. An expanded and interactive version of this figure is available here, Fig 3.

addresses the multiple comparisons issue at the gene set level; therefore, we set the threshold for differentially expressed gene sets to adjusted P-values ≤ 0.05 and the threshold for individual DEGs to P-values ≤ 0.05.

EGSEA identified metabolic pathways for cholesterol and lipid metabolism, the extracellular matrix, inflammation, complement and coagulation immune responses, senescence, and circadian regulation as the pathways significantly altered by serum starvation.

## Cholesterol, lipid, and lipoprotein metabolism pathways

As previously described, serum-deprived ARPE-19 cells respond by upregulating genes in cholesterol-associated pathways [29, 30]. Upregulation is detected by day three and continues increasing throughout the time series. EGSEA ranked the gene set HALLMARK_CHOLES-TEROL_HOMEOSTASIS as the most altered pathway on days 5, 6, and 9. Filtering the list of significantly regulated gene sets to pathways for cholesterol synthesis or pathways for regulating cholesterol levels results in 15 gene sets containing 40 distinct genes (Figs 2 and 3). Of the 25 distinct genes associated with cholesterol biosynthesis (HumanCyc: Encyclopedia of Human Genes and Metabolism), 23 were identified as upregulated in our dataset. Only six of the 40 distinct genes associated with synthesizing and regulating cholesterol levels were downregulated. Three of these genes, *PLTP*, *STARD4*, and *ABCA1*, function in reverse cholesterol transport, the formation of HDL, and its removal from cells. Reducing reverse transport would logically contribute to increased cholesterol levels. Cholesterol levels in the cell are tightly regulated. These gene expression results predict that serum-deprived ARPE-19 cells increase cellular levels of cholesterol over time. Follow-up experiments to detect cholesterol confirm that cholesterol levels increase in these cells [30].

Genes associated with lipid and lipoprotein metabolism were both up and downregulated. Sorting these genes into smaller subsets reveals distinct trends. Genes related to Cytochrome P450, the proteins responsible for the oxidation of retinoic acids, steroids, vitamin D, and cholesterol, were down-regulated. The genes involved in triglyceride biosynthesis were upregulated (see Fig 3C).

The gene expression pattern favors increased lipids and lipoproteins levels (see Fig 3C). Increases in cholesterol and lipids have been described in AMD [1, 11, 13, 14, 17]. Cholesterol and lipids are critical to membrane structures and serve as molecular transporters of visual pigments between the photoreceptors and the RPE [41, 42]. Cholesterol and lipids are major components of the basal deposits and drusen that characterize AMD pathology. Follow-up experiments have demonstrated that serum deprivation of ARPE-19 cells causes increased expression and secretion of unesterified cholesterol and lipids, similar to patterns observed in human AMD eyes [30].

## Extracellular matrix

Among the larger differentially expressed gene sets were those with annotations describing roles in extracellular matrix (ECM) synthesis, maintenance, and function. EGSEA identified

**Basement Membrane-type Collagen DEGs**

| | | Sample Day | | | | | |
|---|---|---|---|---|---|---|---|
| Symbol | GeneName | 1 | 3 | 4 | 5 | 6 | 9 |
| COL4A1 | collagen type IV alpha 1 chain | -1.8 | -3.8 | -2.2 | -1.6 | -1.6 | 1.0 |
| COL4A2 | collagen type IV alpha 2 chain | -1.6 | -3.6 | -2.1 | -1.8 | -1.6 | -1.0 |
| COL4A3 | collagen type IV alpha 3 chain | -1.8 | -1.8 | -1.8 | -1.2 | -1.5 | -1.1 |
| COL4A4 | collagen type IV alpha 4 chain | -1.9 | -2.0 | -1.7 | -1.4 | -1.7 | -1.3 |
| COL4A5 | collagen type IV alpha 5 chain | -2.0 | -1.7 | -1.7 | -1.1 | -1.2 | -1.2 |
| COL4A6 | collagen type IV alpha 6 chain | -2.1 | -1.5 | -3.0 | -1.0 | -1.2 | -1.5 |
| COL8A1 | collagen type VIII alpha 1 chain | -1.8 | -1.4 | -1.1 | 1.1 | -1.1 | -1.4 |
| COL8A2 | collagen type VIII alpha 2 chain | -2.7 | -1.8 | -2.7 | -1.1 | -1.1 | 1.2 |

**Integrin DEGs**

| Symbol | GeneName | 1 | 3 | 4 | 5 | 6 | 9 |
|---|---|---|---|---|---|---|---|
| ITGA1 | integrin subunit alpha 1 | -2.1 | -1.4 | -1.7 | -1.2 | -1.5 | -1.5 |
| ITGA2B | integrin subunit alpha 2b | -2.6 | -2.1 | -2.8 | -1.9 | -2.3 | -2.2 |
| ITGA5 | integrin subunit alpha 5 | -1.6 | -2.2 | -2.1 | -1.6 | -1.5 | -1.2 |
| ITGA6 | integrin subunit alpha 6 | -1.9 | -1.7 | -1.7 | -1.3 | -1.5 | -1.6 |
| ITGAX | integrin subunit alpha X | -3.3 | -3.2 | -3.9 | -2.0 | -2.1 | -2.7 |
| ITGB3 | integrin subunit beta 3 | -2.7 | -2.3 | -2.9 | -2.6 | -2.8 | -3.4 |
| ITGB4 | integrin subunit beta 4 | -1.8 | -1.9 | -2.7 | -1.3 | -1.3 | -1.1 |
| ITGB7 | integrin subunit beta 7 | -1.5 | -1.3 | -2.2 | -3.0 | -2.4 | -3.3 |
| ITGBL1 | integrin subunit beta like 1 | -1.8 | -1.7 | -1.6 | -1.6 | -2.4 | -2.8 |

**Laminins, Nidogens, and Perlecan DEGs**

| Symbol | GeneName | 1 | 3 | 4 | 5 | 6 | 9 |
|---|---|---|---|---|---|---|---|
| HSPG2 | heparan sulfate proteoglycan 2 | -1.9 | -2.2 | -1.5 | -1.0 | -1.2 | -1.2 |
| LAMA3 | laminin subunit alpha 3 | -1.7 | -1.5 | -1.4 | -1.5 | -1.7 | -2.5 |
| LAMA4 | laminin subunit alpha 4 | -2.0 | -2.3 | -2.4 | -2.6 | -3.4 | -4.4 |
| LAMA5 | laminin subunit alpha 5 | -1.7 | -1.8 | -1.9 | -1.0 | -1.2 | -1.0 |
| LAMB2P1 | laminin subunit beta 2 pseudogene 1 | -4.1 | -6.7 | -12.7 | -8.3 | -5.8 | -4.4 |
| LAMB3 | laminin subunit beta 3 | -1.5 | -1.5 | -2.0 | -1.2 | -1.2 | -1.3 |
| LAMC1 | laminin subunit gamma 1 | -2.1 | -2.1 | -2.2 | -1.8 | -2.1 | -2.2 |
| NID1 | nidogen 1 | -2.3 | -2.9 | -1.9 | -2.1 | -2.2 | -2.3 |
| NID2 | nidogen 2 | -4.1 | -4.1 | -7.0 | -3.2 | -3.9 | -3.0 |
| SNED1 | sushi, nidogen and EGF like domains 1 | -2.7 | -2.1 | -1.7 | -1.6 | -2.1 | -2.1 |

**Fibrillar collagen DEGs**

| Symbol | GeneName | 1 | 3 | 4 | 5 | 6 | 9 |
|---|---|---|---|---|---|---|---|
| COL1A1 | collagen type I alpha 1 chain | 1.1 | -2.2 | -2.3 | 2.1 | 1.8 | 7.1 |
| COL3A1 | collagen type III alpha 1 chain | -3.2 | -2.2 | -3.6 | -1.8 | -1.2 | -1.5 |
| COL9A2 | collagen type IX alpha 2 chain | 1.3 | 1.3 | 1.0 | 1.5 | 1.8 | 3.2 |
| COL11A2 | collagen type XI alpha 2 chain | 2.0 | 1.5 | 1.1 | 1.6 | 1.4 | 1.7 |
| COL16A1 | collagen type XVI alpha 1 chain | 1.7 | 1.3 | -1.5 | 1.7 | 1.3 | 3.6 |

Signed fold-change (SFC) for differentially expressed genes (DEG) associated with the extracellular matrix.

Color code:
**Down <-3.0 fold**
Down -1.5 to -3.0 fold
P-value > 0.05
Up 1.5 to 3.0 fold
**Up > 3.0 fold**

**Fig 3. Expression patterns of differentially expressed genes associated with the extracellular matrix.** Highlight display of DEGs for basement membrane collagens, integrins, laminins, nidogens, perlecan, and fibrillar collagens. Genes are represented in the rows. The SFC for each sample day is represented by squares color-coded for the magnitude of the fold change. Orange squares represent down-regulated expression, blue squares represent up-regulated expression, and grey squares have P-values greater than 0.05.

18 gene sets that contained 221 distinct DEGs. The gene expression patterns indicated remodeling of the ECM. Generally, genes associated with basal membranes were downregulated early in the time series. At the end of the time series, on days 6 and 9, transcription related to a fibrotic extracellular matrix was increased (Fig 3, see Fig 4).

The differentially expressed gene sets included six sets identified by Naba et al. in the Matrisome project [43, 44]. The Matrisome project is a combination *in silico/in vivo* effort to identify

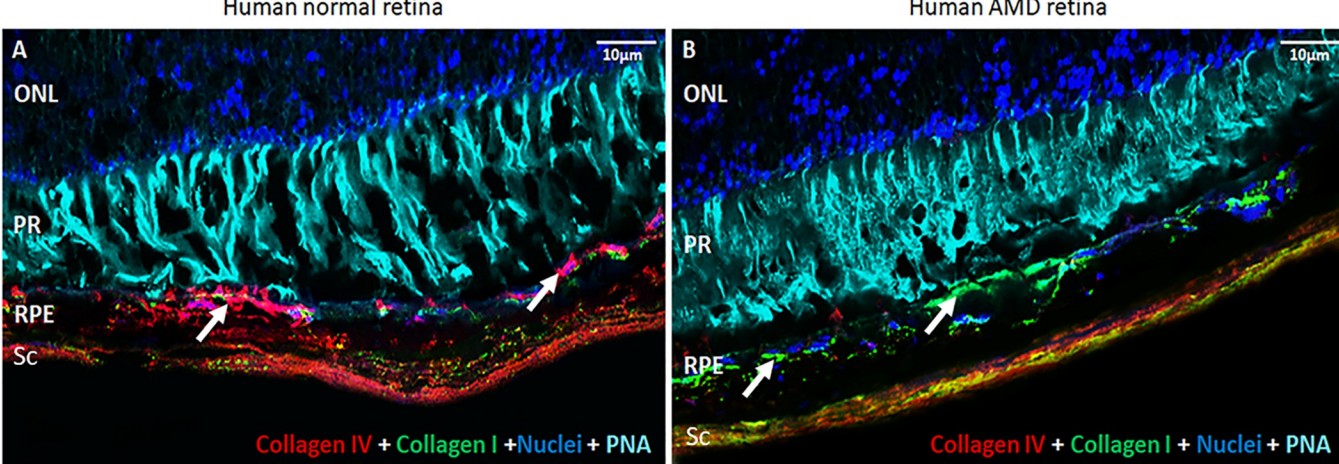

**Fig 4. Collagen type I vs. collagen type IV in normal and AMD human retina.** Human retinas probed with collagen type I, collagen type IV, DAPI (nuclei), and PNA (detects cone photoreceptors). A. normal retina. B. AMD retina. ONL- outer nuclear layer, PR–photoreceptors, RPE–retinal pigment epithelium, Sc–sclera. Arrows indicate collagen in Bruch's membrane.

and categorize the proteome of the ECM. The proteins identified by the project are categorized into core matrisome and matrisome associated. Both core matrisome and matrisome associated gene sets were identified by EGSEA in our data. The meshwork of proteins between cells and the specialized basement membranes that underlie epithelial cells constitute the core matrisome. During the time series, gene sets for the basement membrane and ECM glycoproteins were differentially expressed. Matrisome associated proteins, transmembrane proteins that allow cells to adhere to the matrix, ECM regulators, and secreted factors were also differentially expressed (see Fig 4).

Genes coding for the basement membrane were quickly and consistently downregulated. Collagen type IV, laminins, nidogens, and *HSPG2* (perlecan); major components of Bruch's membrane, a specialized basement membrane and basal lamina deposition and drusen formation site; were down-regulated. Integrins, which modulate cell-cell binding via attachment to laminin, were also downregulated [45–47]. Near the end of the time series, gene expression for a different, more fibrous extracellular matrix was upregulated (Fig 3, see Fig 4, see Fig 5B) [48]. Immunohistochemical staining of human eye sections comparing eyes with drusen and diagnosed AMD to similarly aged disease-free eyes demonstrate a shift from predominantly Collagen type IV in the disease-free eyes to Collagen type I in AMD eyes (Fig 4).

Matrix metallopeptidase genes (MMPs) and ADAM metallopeptidase (ADAM) or ADAM metallopeptidase with thrombospondin (ADAMTS) genes, the two major types of matrisome ECM regulators responsible for degrading and remodeling the ECM [49, 50], were differentially expressed by serum-deprived ARPE-19 cells (see Fig 4C). Especially notable is MMP2, a matrix metallopeptidase known to play a role in AMD pathology [19, 51]. Genes that modulate the MMPs, ADAMS, and ADAMTS, such as TIMP metallopeptidase inhibitor 3 (*TIMP3*), were not differentially expressed during the time series.

The ECM of the RPE composes and supports the retina facing half of the specialized basement membrane of the eye, the Bruch's membrane [20]. The regulation of pathways associated with the extracellular matrix seen in this data indicates a change in the nature of the ECM during serum deprivation, transitioning from a basement membrane-type ECM to an ECM characteristic of fibrotic tissues, with increases in fibrillar collagens [52]. The change in the

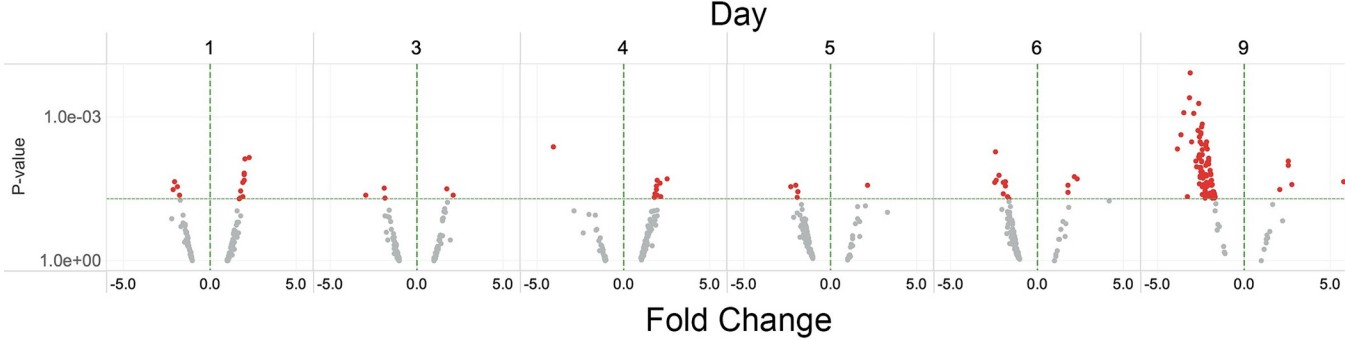

**Fig 5. Volcano plots of cell cycle genes.** Scatter plots of RNASeq DESeq results, P-value vs. Fold Change (volcano plots),for each sample day arranged chronologically and filtered to genes involved in the cell cycle. Each point represents a gene. The Fold change plotted on the X-axis is the SFC. The P-value is plotted on the Y-axis. Color represents the P-value; red points have P-values ≤ 0.05, and grey points have P-values > 0.05. The X-axis reference line is SFC = 0. The Y-axis reference line is P-value = 0.05. While the Y-axis is rendered in log scale, the actual p-value labels are given along the axis. Fig 5 presents an expanded and interactive version of this figure.

expression of matrix remodeling proteinases, MMPs, ADAMs, and ADAMTSs is consistent with a shift from a basement membrane ECM to a fibrous ECM.

A more fibrous matrix impedes the movement of secreted molecules. Combined with the increase in cholesterol, this could create an environment that results in the accumulation of basal deposits and, eventually, drusen formation (see Figs 4 and 5B). This, in turn, might lead to the recruitment of microglia and macrophages, resulting in a shift to activation of the innate immune response and the complement cascade [51, 52].

Among the DEGs not included in any of the Gene Sets is *amelotin* (*AMTN*) (see Fig 7), which codes for a protein responsible for the calcification of hydroxyapatite in tooth enamel [53–55]. The induction of *AMTN* expression is an important discovery since calcified deposits or nodules are markers for the progression of dry AMD or geographic atrophy (GA) [56–58]. *AMTN* expression was increased 58-fold by day 9. Amelotin belongs to the core matrisome category. Follow-up experiments have demonstrated that *AMTN* is present in AMD eyes with GA associated with hydroxyapatite deposits in drusen. Work to characterize amelotin and understand its function in the eye is ongoing [59–62].

## Inflammation

EGSEA detected significant differential expression of the Hallmark Inflammatory Response gene set on days 3, 5, 6, and 9. There were 45 distinct DEGs from the Hallmark Inflammatory Response gene set. DEGs were both up and downregulated during the time series. The largest number of differentially expressed genes appeared on day 9. The DEGs varied considerably during the time series, with three-quarters meeting the P-value cut-off on only one or two of the time points. Despite the variability of the individual genes, Inflammatory pathways have been identified by EGSEA, which employs multiple distinct statistical methods for identifying differentially expressed gene sets. Heightened variability of genes associated with inflammation characterizes the state known as parainflammation. Parainflammation is believed to be a precursor of macular degeneration [63–65].

The nature of inflammatory and immune homeostasis, with both stimulated and repressed genes, makes regulation patterns more difficult to detect. One hundred fourteen distinct genes in this dataset are annotated with Biological Process GO terms for inflammation. Sorting these genes by GO term identifies multiple DEGs for inflammation, positive and negative regulation of inflammation, and acute inflammation, but only a single DEG was annotated for chronic

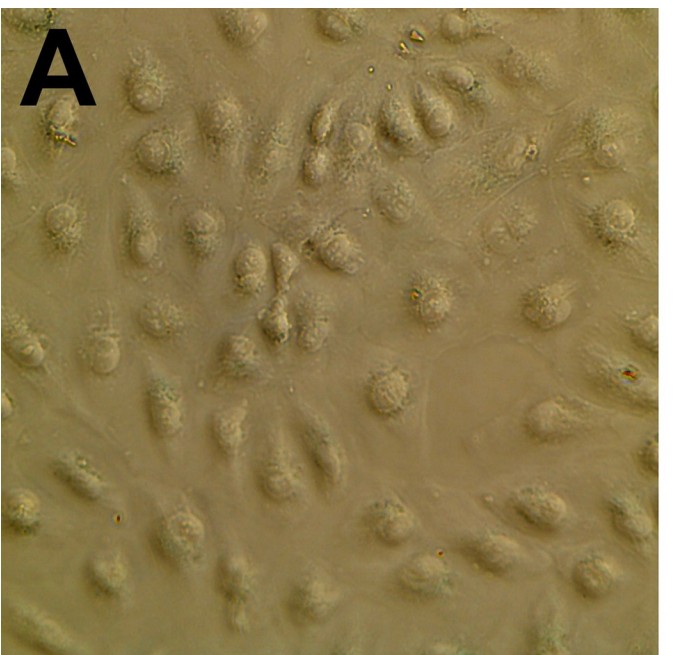
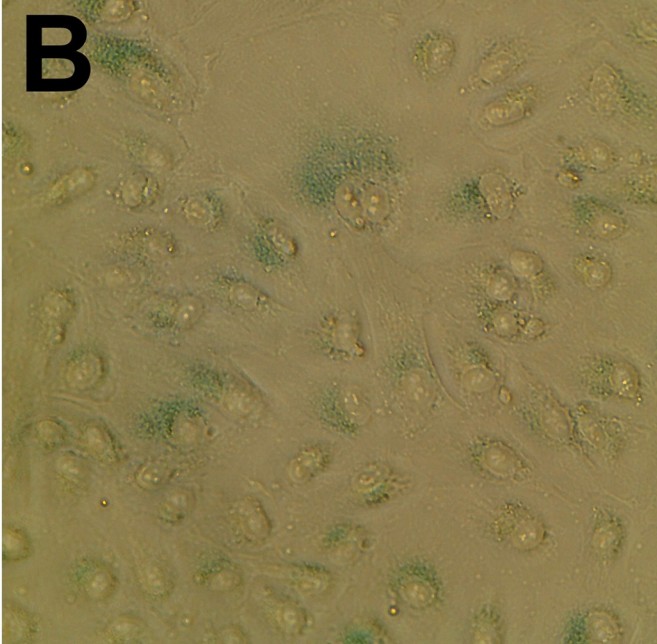

**Fig 6. ß-galactosidase activity in serum-deprived ARPE-19 cells.** A. ARPE-19 cells cultured in serum-supplemented media stained for ß-galactosidase activity. B. ARPE-19 cells cultured in serum-free media for nine days and stained for ß-galactosidase activity. The ß-galactosidase substrate turns blue in the presence of ß-galactosidase activity.

inflammatory response, implying that these cells do not directly trigger chronic inflammation (see Fig 6A–6C). Instead, the macrophages and microglia that infiltrate the RPE *in vivo* in response to oxidative stress may be responsible for producing the chronic inflammatory response.

## Complement and coagulation

EGSEA identified three large gene sets for complement and coagulation: Hallmark Complement, Hallmark Coagulation, and Wikipathway Complement and Coagulation Cascades. These sets contained 54 unique DEGs. Members of these sets overlap substantially. Hallmark Complement annotates 36 genes, Hallmark Coagulation annotates 31 genes, and Complement and Coagulation Cascade annotates 15 genes with 7 genes annotated by all 3 sets. Notably, genes for the receptors for C3A and C5A (*C3AR1* and *C5AR1*) that participate in forming membrane attack complex pores (MAC) were upregulated. Downregulated genes (37 total) outnumbered upregulated genes (17 total) (see Fig 6D).

A balance of stimulation and inhibition characterizes complement and coagulation pathways in the eye. Serum deprivation of ARPE-19 cells results in both up and downregulation of the genes from these pathways (see Fig 6). Activation of coagulation, an offshoot of the complement cascade, results in the formation of fibrous tissue [66]. The activity of the complement and coagulation pathways as early as day one is likely to be associated with the changes in the extracellular matrix from basement membrane to fibrous type.

## Senescence

The final response of these cells is to enter into a state of senescence. Senescence shuts down cellular functions without leading to immediate cell death. It is a state that preserves the RPE

cells but fails to serve the metabolic needs of the photoreceptors. Senescence is a state that favors the development of age-related diseases and is believed to have a role in AMD [9, 13, 23, 67]. Senescence is characterized by changes in the extracellular matrix, the expression of matrix-degrading enzymes, inflammatory cytokines, and profound suppression of the cell cycle.

In serum-deprived ARPE-19 cells at the end of the time course, there is a significant down-regulation of genes involved in the cell cycle, mitosis, and cell division, and upregulation of matrix metalloproteinases and fibrosis-associated collagen genes. Gene Sets associated with the cell cycle and DNA replication contained 112 distinct genes that meet the P-value criteria. These genes and sets had a distinctive expression pattern: predominately upregulated from days 1 through 4, then, on day 5, the dynamic reversed when downregulated genes predominated with a sharp spike of downregulated genes on day 9 (Fig 5, see Fig 5A). Combined with the ECM results, this suggests that the cells are entering senescence.

To determine whether serum-deprived ARPE-19 cells are entering senescence, we assayed the cells for ß-galactosidase activity, an established assay for senescence. Serum-supplemented cells were negative for ß-galactosidase activity. Cells cultured for nine days in serum-free media were positive for ß-galactosidase activity (Fig 6).

### Circadian regulation

There were 26 DEGs annotated with GO Biological Process terms that describe functions responsible for establishing and maintaining circadian rhythms (Fig 7 and 7A). There were 19 downregulated genes and nine upregulated genes. Parsing the descriptions of the gene's functions implied a decrease in the regulation of circadian rhythm. Given the estimates of the number of genes subject to circadian control and their roles in maintaining visual cycle proteins, disrupting the circadian regulatory pathways could have a widespread negative impact on the maintenance and health of the retina.

### Unannotated differentially expressed genes

Gene annotation is incomplete; there are differentially expressed genes without annotations in the MSigDB or GSdb (see Fig 1). Genes that lack annotations do not contribute to the gene set analysis. We selected and examined unannotated DEGs to determine whether they could be manually incorporated into this study (see Fig 7). The list of unannotated genes consists of genes with fold change less than -5 or greater than 5. This list was further restricted to genes with raw expression counts greater than 10 to manage the list's size and reduce false positives (see Fig 7C).

Filtering the DEGs with these criteria yielded a list of 71 genes, 37 upregulated and 34 downregulated (see Fig 7B). Despite their absence from MSigDB and GSdb, 46 of these genes are annotated by GO, leaving 25 DEGs with large fold changes that are entirely unannotated. Examining the GO annotations accounted for some of these genes, placing them into one of the major categories identified by EGSEA (see Fig 7C). *Aquaporin 5* (*AQP5*), an upregulated gene, shares the GO:BP annotation Odontogenesis with AMTN. Other MSigDB/GSdb unannotated genes included four downregulated genes for keratin formation, all from the hair subtype, and four downregulated pregnancy-specific beta-1-glycoprotein genes (see Fig 7C).

### Conclusions

Our analysis of the time series of serum-starved ARPE-19 cells found that these RPE cells respond by up and down regulating pathways responsible for the etiology of AMD. APRE-19 cells respond to serum starvation by modulating the gene expression of cholesterol, lipid,

## Genes with Circadian Function Annotations and P-value < 0.05

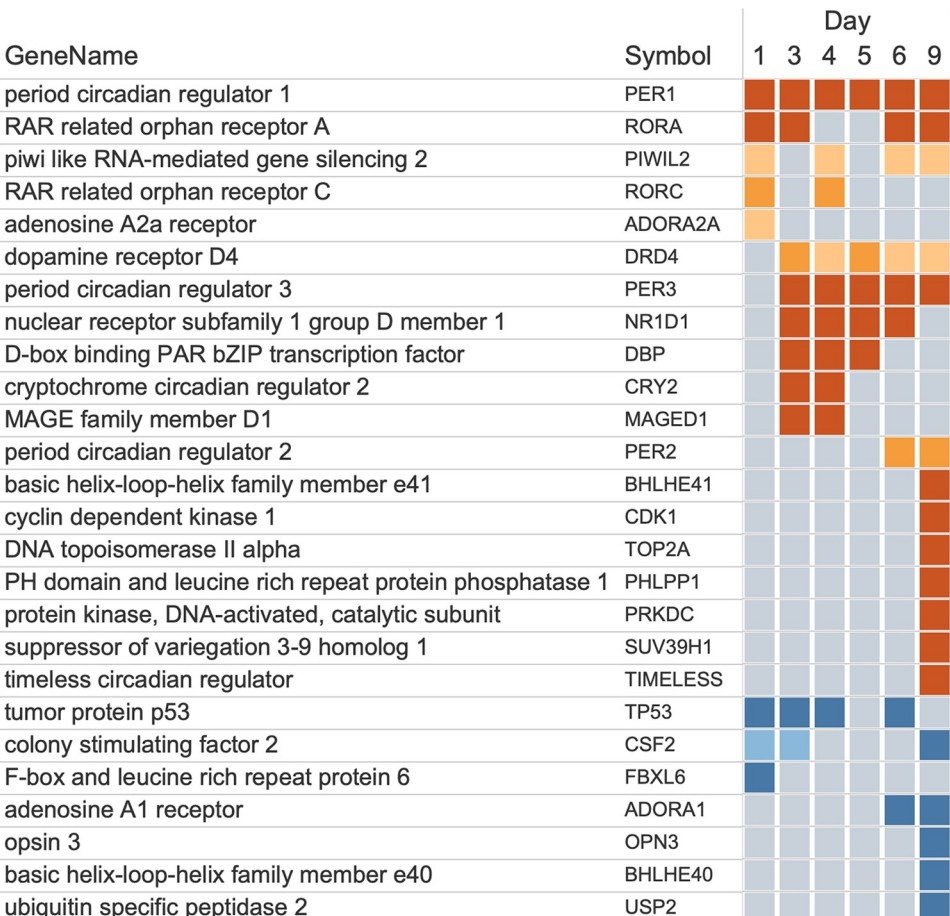

| GeneName | Symbol | Day 1 | 3 | 4 | 5 | 6 | 9 |
|---|---|---|---|---|---|---|---|
| period circadian regulator 1 | PER1 | | | | | | |
| RAR related orphan receptor A | RORA | | | | | | |
| piwi like RNA-mediated gene silencing 2 | PIWIL2 | | | | | | |
| RAR related orphan receptor C | RORC | | | | | | |
| adenosine A2a receptor | ADORA2A | | | | | | |
| dopamine receptor D4 | DRD4 | | | | | | |
| period circadian regulator 3 | PER3 | | | | | | |
| nuclear receptor subfamily 1 group D member 1 | NR1D1 | | | | | | |
| D-box binding PAR bZIP transcription factor | DBP | | | | | | |
| cryptochrome circadian regulator 2 | CRY2 | | | | | | |
| MAGE family member D1 | MAGED1 | | | | | | |
| period circadian regulator 2 | PER2 | | | | | | |
| basic helix-loop-helix family member e41 | BHLHE41 | | | | | | |
| cyclin dependent kinase 1 | CDK1 | | | | | | |
| DNA topoisomerase II alpha | TOP2A | | | | | | |
| PH domain and leucine rich repeat protein phosphatase 1 | PHLPP1 | | | | | | |
| protein kinase, DNA-activated, catalytic subunit | PRKDC | | | | | | |
| suppressor of variegation 3-9 homolog 1 | SUV39H1 | | | | | | |
| timeless circadian regulator | TIMELESS | | | | | | |
| tumor protein p53 | TP53 | | | | | | |
| colony stimulating factor 2 | CSF2 | | | | | | |
| F-box and leucine rich repeat protein 6 | FBXL6 | | | | | | |
| adenosine A1 receptor | ADORA1 | | | | | | |
| opsin 3 | OPN3 | | | | | | |
| basic helix-loop-helix family member e40 | BHLHE40 | | | | | | |
| ubiquitin specific peptidase 2 | USP2 | | | | | | |

Expression Levels
- Down-regulated High expression
- Down-regulated Mid expression
- Down-regulated Low expression
- Unchaged
- Up-regulated Mid expression
- Up-regulated High expression

**Fig 7. Genes annotated with circadian functions and their expression patterns.** The list of genes and their symbols. Each square represents the gene expression level in raw counts and the change direction. Orange squares represent downregulation, and blue squares represent upregulation. The darker shades of color represent higher expression levels (see Fig 7A).

extracellular matrix, complement and coagulation, inflammation, senescence, and circadian rhythm pathways. Examining the timing and specifics of differential gene expression using the systems biology framework of AMD has increased our understanding of the pathology of AMD. We have also identified a gene, *AMTN*, that may play a critical role in the pathology of geographic atrophy [59–62].

The significant response of so many of the biological systems implicated in the development and progression of AMD leads us to contend that serum deprivation of ARPE-19 cells is an informative model for identifying mechanisms underlying the aging RPE's response to the

collapse of the underlying choroid [29, 68]. This is a simple model with weaknesses and strengths; physiological conditions are more complex than culture conditions, but these cells maintain their epithelial identity [29]. The system's simplicity allows easy and rapid investigation of the underlying programming of the RPE. Gene expression studies alone do not provide a complete picture of the contents and milieu of these cells. Confirmation and further experiments will be required, and more elaborate cell cultures or animal models should provide additional insight. Indeed, several follow-up projects from this work have employed more sophisticated or elaborate model systems[30, 59–62].

Our study developed a narrative explanation for APRE-19 cell response to serum deprivation. By performing multiple analyses using a variety of gene set collections and incorporating as much gene information as possible, we expanded our understanding of the response of epithelial cells to conditions of serum deprivation. Our insights generated hypotheses and directions for further study, and the interactive visualizations continue to serve as a resource for further research.

## Supporting information

**S1 Table. Transcript count of RPE marker genes.** The normalized mean RNASeq count of RPE marker genes (identified in Reyes et al [31]) in ARPE-19 cells. The genes are identified by official HUGO symbol. Counts, for each sample day, represent the mean of 3 biological replicate samples measured in triplicate. Columns: *EntrezID*, Symbol, Gene Name, Counts Day0, Day1, Day3, Day4, Day5, Day6, Day9.
(XLSX)

**S2 Table. DESeq results.** The results of the DESeq Analysis. Columns: Sample, *EntrezID*, Symbol, Day 0 Mean, SampleDayMean, FoldChange, log2FoldChange, pval, padj.
(TXT)

**S3 Table. Significant gene set results from EGSEA.** The significant gene sets returned by EGSEA analysis of serum-deprived ARPE-19 cell differentially expressed genes. The table contains the results from two runs of the EGSEA algorithm, one specifying the Hallmark gene set annotation collection, the other specifying the C2 division of MSigDB, which includes Bio-Carta, KEGG, and Reactome. The table consists of columns for Gene Set Name, Rank, P-value, adjusted P-value, the general direction (increased or decreased) of the gene expression, and Sample Day.
(XLSX)

**S1 Fig. Interactive figure information.** Information and links for the interactive figures that accompany this paper.
(DOCX)

## Author Contributions

**Conceptualization:** Katherine M. Peterson, Sanghamitra Mishra, Graeme Wistow.

**Data curation:** Katherine M. Peterson, Esther Asaki, John I. Powell, Yiwen He, Alan E. Berger.

**Formal analysis:** Katherine M. Peterson, Esther Asaki, John I. Powell, Yiwen He, Alan E. Berger.

**Investigation:** Katherine M. Peterson, Sanghamitra Mishra, Esther Asaki, John I. Powell, Yiwen He, Alan E. Berger, Dinusha Rajapakse.

**Project administration:** Katherine M. Peterson, Graeme Wistow.

**Supervision:** Graeme Wistow.

**Visualization:** Katherine M. Peterson, Dinusha Rajapakse.

**Writing – original draft:** Katherine M. Peterson, Sanghamitra Mishra, Esther Asaki, Alan E. Berger, Dinusha Rajapakse.

**Writing – review & editing:** Katherine M. Peterson, Sanghamitra Mishra, Esther Asaki, John I. Powell, Yiwen He, Alan E. Berger, Dinusha Rajapakse, Graeme Wistow.

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
