## [Decision Letter · Decision Letter 0]

18 Dec 2023

PONE-D-23-33089Serum-deprivation response of ARPE-19 cells; expression patterns relevant to age-related macular degeneration.PLOS ONE

Dear Dr. Peterson,

Thank you for submitting your manuscript to PLOS ONE. After careful consideration, we feel that it has merit but does not fully meet PLOS ONE’s publication criteria as it currently stands. Therefore, we invite you to submit a revised version of the manuscript that addresses the points raised during the review process.

We look forward to receiving your revised manuscript.

Kind regards,

Steven Barnes

Academic Editor

PLOS ONE

3. We are unable to open your Supporting Information file [S1_Table.txt ]. Please kindly revise as necessary and re-upload.

Additional Editor Comments:

After carefully considering Reviewer 1's comments, and carefully reading through the manuscript myself, I fully concur with the Reviewer's comments and request that you proceed to address the Reviewer's comments.

Reviewers' comments:

Reviewer's Responses to Questions

**Comments to the Author**

1. Is the manuscript technically sound, and do the data support the conclusions?

Reviewer #1: Partly

2. Has the statistical analysis been performed appropriately and rigorously? 

Reviewer #1: Yes

3. Have the authors made all data underlying the findings in their manuscript fully available?

Reviewer #1: Yes

4. Is the manuscript presented in an intelligible fashion and written in standard English?

Reviewer #1: Yes

5. Review Comments to the Author

Reviewer #1: The manuscript employed ARPE-19 cells under serum deprivation to model RPE in AMD and analyzed gene expression changes and signaling pathways via RNA sequencing and GSEA analysis. It reported alterations in AMD-associated pathways in serum-deprived ARPE-19 cells.

Major comments:

1. ARPE-19 cells, due to their low expression of RPE signature genes and poor transepithelial electrical resistance scores, do not accurately represent RPE cells. This study utilized cells at 80% confluence, which even lacked the critical RPE polarity. Models such as iPSC-RPE or primary human RPE cells offer more accurate in vitro representations for these analyses. Thus, the limitations of the ARPE-19 cell model significantly undermine the study's relevance to AMD.

2. The manuscript could benefit from improvements in readability and structure:

Figures should be presented sequentially as they are referred to in the text. For instance, Figure S5B, which illustrates ECM genes, could be incorporated into Figure S4. Moreover, Figure 4's discussion (lines 291-294) precedes that of Figure 3; hence, it would be logical to introduce Figure 4 earlier in the text to maintain coherence.

The supplementary material contains a lot of information that requires frequent access. Consider integrating crucial figures into the main text. Duplication of figures in both the main and supplementary sections leads to redundancy.

Explicitly connect figure parts to the corresponding textual statements. For example, after the statement in lines 254-255 regarding triglyceride biosynthesis, directly refer to the specific panel in Figure S3 that supports this claim to guide the reader effectively.

Minor comments:

The claim in lines 262-263 regarding serum deprivation effects on ARPE-19 cells needs direct support from Figure S3, which currently appears to be missing.

In discussing the extracellular matrix, a heatmap illustrating the fold changes of the genes mentioned in the main text, such as MMP2 and TIMP3 would enhance comprehension.

The manuscript does not clearly identify the genes related to fibrotic extracellular matrix within the figures.

The abbreviations ADAM and ADAMTS should be spelled out upon first use to ensure clarity for all readers.

The statement in line 379 about senescence in the context of AMD cites a study using ARPE-19 cell models, which is insufficient to substantiate the role of senescence in AMD.

Immunostaining figures should use arrows to delineate the RPE layer from the choroid, so that the staining signals can be distinguished. In addition, the number of human donor eyes utilized for staining should be disclosed.

For volcano plot figures and legends:

Axis labels need clarity; if P-value is log-scaled, the figure legend should specify it, for example -log10(P-value).

The reference line labeled "0.05" is inconsistent with a log scale. It should reflect the corresponding -log10(0.05) value if the Y-axis is indeed log-scaled.

Figure 4 lacks tick marks on axes.

In Figure 7, a color scale bar could more efficiently convey expression levels, streamlining the figure.

6. PLOS authors have the option to publish the peer review history of their article (what does this mean?). If published, this will include your full peer review and any attached files.

Reviewer #1: No

---

## [Author Response · Author response to Decision Letter 0]

25 Jan 2024

We have uploaded a Response to reviewers document.

---

## [Editor Report · Decision Letter 1]

29 Jan 2024

Serum-deprivation response of ARPE-19 cells; expression patterns relevant to age-related macular degeneration.

PONE-D-23-33089R1

Dear Dr. Peterson,

We’re pleased to inform you that your manuscript has been judged scientifically suitable for publication and will be formally accepted for publication once it meets all outstanding technical requirements.

Kind regards,

Steven Barnes

Academic Editor

PLOS ONE

Additional Editor Comments (optional):

Thank you for thoroughly addressing the sole referee's concerns. I have carefully reviewed your responses and changes to the manuscript and I find your response to be sufficient to merit acceptance at this time.

We regret the long time it took for this manuscript to reach this final, positive decision.
---

## [Editor Report · Acceptance letter]

27 Feb 2024

PONE-D-23-33089R1 

PLOS ONE

Dear Dr. Peterson, 

I'm pleased to inform you that your manuscript has been deemed suitable for publication in PLOS ONE. Congratulations! Your manuscript is now being handed over to our production team.

Kind regards, 

on behalf of

Dr. Steven Barnes 

Academic Editor

PLOS ONE